# Optimal Motion Control of a Capsule Endoscope in the Stomach Utilizing a Magnetic Navigation System with Dual Permanent Magnets

**DOI:** 10.3390/mi15081032

**Published:** 2024-08-14

**Authors:** Suhong Bae, Junhyoung Kwon, Jongyul Kim, Gunhee Jang

**Affiliations:** 1Department of Mechanical Convergence Engineering, Hanyang University, Seoul 04763, Republic of Korea; fjur564@hanyang.ac.kr (S.B.); shqpffhtm@hanyang.ac.kr (J.K.); rlawhddbf@hanyang.ac.kr (J.K.); 2Department of Mechanical Engineering, Hanyang University, Seoul 04763, Republic of Korea

**Keywords:** capsule endoscope, magnetic field, magnetic force, magnetic navigation system, optimization problem, permanent magnet, point dipole model

## Abstract

We propose a method to control the motion of a capsule endoscope (CE) in the stomach utilizing either a single external permanent magnet (EPM) or dual EPMs to extend the examination of the upper gastrointestinal tract. When utilizing the conventional magnetic navigational system (MNS) with a single EPM to generate tilting and rotational motions of the CE, undesired translational motion of the CE may prevent accurate examination. We analyzed the motion of the CE by calculating the magnetic torque and magnetic force applied to the CE using the point-dipole approximation model. Using the proposed model, we propose a method to determine the optimal position and orientation of the EPM to generate tilting and rotational motions without undesired translational motion of the CE. Furthermore, we optimized the weight of dual EPMs to develop a lightweight MNS. We prototyped the proposed MNS and experimentally verified that the developed MNS can generate tilting and rotational motions of the CE without any translational motion.

## 1. Introduction

The incidence of gastrointestinal (GI) disease in modern society is gradually increasing due to stress, irregular eating habits, etc. [1,2,3]. Sexton et al. showed that patients with stage 3 gastric cancer had a death rate between 50 and 82% within 5 years after surgery [4]. For early diagnosis and treatment of GI disease, wired endoscopy is becoming popular. Most wired endoscopies are performed under general anesthesia because the wired tube of the endoscopic instrument running through the esophagus can be uncomfortable and painful for the patient. However, general anesthesia requires several hours of recovery before the patient can return to normal life. Sometimes, the wired endoscope may cause gastrointestinal bleeding and perforation. Examination of the small intestine requires a high level of expertise. To overcome these issues, the capsule endoscope (CE), which is a small, pill-shaped device that can be swallowed easily, has been developed to examine the small intestine [5]. The CE moves passively through the gastrointestinal tract by peristalsis and is equipped with a CMOS camera that can capture images of the examination area at a rate of approximately 0.5 to 38 frames per second, providing continuous imaging of the small intestine [6]. However, it cannot examine the large hollow form of the stomach because the motion of the CE cannot be actively controlled [7]. Figure 1 shows the three types of motion that the CE must perform to examine the stomach: translational motion to move the CE to a target location; tilting motion to modify the shooting angle of the CE; and rotational motion to rotate the tilted CE to scan around a specific axis to inspect the gastrointestinal tract.

There has been much research with several different structures of CEs to develop an effective active control for the motion of the CE. Some researchers have proposed a CE with propellers, legs, anchor, balloon, inchworm-like locomotion or gear system to control the motion actively [8,9,10,11,12,13,14,15]. However, because these devices have a hydraulic or electrical driving module located inside the capsule, they are larger than commercial capsule endoscopes (with an average length of 27.6 mm) which makes it difficult for them to be swallowed through the human esophagus [7,16,17]. To overcome these issues, a CE with a permanent magnet (PM) inside has been developed. These magnets enable the CE to be driven using the magnetic forces and torques generated by external magnetic fields. Such magnetic fields can be generated using a magnetic navigation system (MNS), which is classified into electromagnet or PM types. The electromagnet-type MNS has attracted a great deal of interest because it can produce the desired magnetic fields by changing the current applied to the coils of the electromagnets [18,19,20,21,22,23,24,25,26,27]. Despite these advantages, the magnetic field generated by electromagnets is relatively weaker than that produced by a PM. Additionally, it requires a separate power supply to apply current, and the heating issue of electromagnets makes it challenging for prolonged use in actual examination environments. On the other hand, PM-type MNSs are free from the aforementioned issues. By mechanically controlling the position and orientation of the PM, it is possible to control the motion of the CE [26,27,28,29,30,31,32]. Several PM-type MNSs have been developed and are commercially available [33,34,35,36,37,38]. Commercialized PM-type MNSs utilize a single external PM (EPM). Several researchers have studied the optimal position and orientation control of CE using EPM. Arthur et al. developed an optimal control by approximating the EPM attached to the robot arm and the capsule endoscope with a point dipole model [39]. Mohammed et al. investigated a position control by approximating the capsule endoscope with a point dipole model [40]. In addition, Taddese et al. analyzed the magnetic field of the EPM and the capsule endoscope with a point dipole model and FEM [41]. To control the orientation of the EPM to generate the tilting motion of the CE, as shown in Figure 2, both magnetic torque and horizontal magnetic force are generated and applied to the CE. In this case, if the friction force acting between the CE and the contact surface of the GI tract is not sufficient, undesired translational motion of the CE can hinder precise motion control.

Moreover, this magnetic force prevents the CE from rotating about a single contact point during the generation of its rotational motion. This magnetic force moves the contact point in a radial direction, making it difficult to capture accurate images of the target area. When undesired motion of the CE occurs, additional movement of the EPM is required to move the CE back to its original position, which prolongs the examination time. This undesired motion of the CE occurs when the horizontal magnetic force is greater than the friction force. Increasing the volume of the EPM enhances the frictional force but also increases the overall size and weight of the MNS. In addition, increasing the volume of the EPM increases the horizontal magnetic force, which makes it difficult to avoid this undesired translational motion. On the other hand, an MNS composed of dual EPMs provides diverse combinations of position and orientation of the EPMs to generate the required magnetic flux density and forces. Kim et al. proposed an optimization problem that determines the magnetization direction of two EPMs to generate the magnetic field required to drive the CE’s tilting motion with minimized horizontal magnetic force [42]. However, they did not consider the dynamics of the CE, which could result in undesired translational motion. Their MNS had limited control freedom because of the fixed relative positions of the EPMs, allowing change in only the magnetization direction of the two EPMs. Additionally, the distance between the CE and EPMs needed to be around 60 mm for effective control, making it difficult for practical clinical application.

In this study, we propose a method to control the motion of the CE using a lightweight MNS with dual EPMs. Using the point-dipole approximation model, we determine the optimal position and orientation of the EPMs to generate tilting and rotational motions without undesired translational motion of the CE. Furthermore, we optimize the weight of each EPM to develop a lightweight MNS. We prototype the proposed MNS and experimentally verify that the developed MNS can effectively generate tilting and rotational motions of the CE without undesired translational motion.

## 2. Materials and Methods

### 2.1. Optimal Motion Contorol of a CE

#### 2.1.1. Point Dipole Analysis of a CE and EPMs

We utilize the point-dipole approximation model to mathematically analyze the magnetic flux density generated by n EPMs and the magnetic force applied to the CE as follows [43]:(1)$$B_{EPM}=\sum_{k=1}^{n}\frac{\mu_{0}}{4\pi }(\frac{3r_{k}\left(m_{EPM,k}\cdot r_{k}\right)}{{r_{k}}^{5}}-\frac{m_{EPM,k}}{{r_{k}}^{3}})$$
(2)$$F_{CE}=\sum_{k=1}^{n}\frac{3\mu_{0}}{4\pi {r_{k}}^{5}}\left\{\left(m_{EPM,k}\cdot r_{k}\right)m_{CE}+\left(m_{CE}\cdot r_{k}\right)m_{EPM,k}+\left(m_{EPM,k}\cdot m_{CE}\right)r_{k}-\frac{5\left(m_{EPM,k}\cdot r_{k}\right)\left(m_{CE}\cdot r_{k}\right)}{{r_{k}}^{2}}\right\}$$
where n is the number of EPMs, $r_{k}$ is the displacement vector from the k-th EPM to the PM of the CE, $\mu_{0}$ is the permeability of the vacuum ($4\pi \times 10^{-7}$ H/m), and $m_{EPM,k}$ and $m_{CE}$ are the magnetic dipole moments of the k-th EPM and PM of the CE, respectively. The volume of the PM (V) and residual magnetic flux density of the PM ($B_{r}$) are used to compute the magnetic dipole moment (m) as follows:(3)$$m=\frac{B_{r}V}{\mu_{0}}$$
Using (1) to (3), we can design EPMs capable of generating the magnetic flux density and force required to control the motion of the CE.

During the examination, it is essential to position the CE at the upper part of the stomach while ensuring that the EPM does not come into contact with the patient. In this study, we considered the case where the CE is positioned in the upper part of the stomach not only because it is the most difficult case, but also because we can apply the same approach when it is located in the lower part of the stomach. We also assumed that the waist thickness of men between 45 and 49 years old in a supine position is between 200 and 300 mm, based on the human body dimension data provided by the Korean Agency for Technology and Standards [44]. Since the location of the stomach varies from person to person, in this study, the maximum vertical distance between the EPM and CE was set to 300 mm.

#### 2.1.2. Force Analysis of a CE

The motion of the CE used in GI examination is generated by controlling the position and orientation of the EPM. Variations in the orientation and position of the EPM change $m_{EPM}$ and r, resulting in changes in the magnetic flux density $B_{EPM}$ at the location of the CE and the magnetic force $F_{CE}$ exerted on the CE. Translational motion of the CE occurs until the sum of all horizontal forces acting on the CE is equal to zero, and tilting motion of the CE occurs until the sum of all torques applied to the CE is equal to zero.

Figure 3 is the free body diagram of the CE, which shows the horizontal and vertical magnetic forces exerted on the CE by the EPMs $F_{x}$ and $F_{z}$, the gravitational force $F_{g}$, friction force $F_{f}$, normal force $F_{N}$, and magnetic torque $T_{B}$. To position the CE in the upper part of the stomach, the vertical magnetic force must be greater than the weight of the CE, as follows:(4)$$F_{z}\geq F_{g}$$

The CE will be under the translational motion until the horizontal magnetic force is greater than the static friction force between the CE and the contact surface, as follows:(5)$$F_{x}-F_{f}=F_{x}-{\mu_{s}(F}_{z}-F_{g})>0$$
where $\mu_{s}$ is the maximum static frictional coefficient between the CE and the contact surface. Then, its translational motion will stop when the static friction force is equal to or greater than the horizontal magnetic force, as follows:(6)$$F_{f}-F_{x}={\mu_{s}(F}_{z}-F_{g})-F_{x}\geq 0$$

When no external forces are applied, the PM of the CE aligns along the direction of the external magnetic flux density $B_{desired}$ generated by the EPMs. However, various external forces are applied to the CE, as shown in Figure 3, and the CE is tilted with the aligning angle when the magnetic torque $T_{B}$ in Equation (7) is equal to the torque $T_{F}$ in Equation (8), which is generated by external forces applied to the CE.
(7)$$T_{B}=m_{CE}\times B_{EPM}$$
(8)$$T_{F}=r_{CE}\times (F_{z}+F_{g}+F_{x})$$
where $r_{CE}$ is the displacement vector from the contact point of the CE to its center of mass. Due to the torque exerted by external forces, there is a difference between the desired tilt angle of the CE and the actual tilt angle, as follows:(9)$$\theta_{\mathrm{error}}=\left|\cos^{-1}\left(\frac{B_{\mathrm{EPM}}\cdot B_{\mathrm{desired}}}{\left\|B_{\mathrm{EPM}}\right\|\cdot \left\|B_{\mathrm{desired}}\right\|}\right)\right|$$
to align the CE in the desired direction, $\theta_{error}$ must be minimized.

#### 2.1.3. Optimal Control of the EPMs for Tilting and Rotational Motion of the CE

Figure 4 shows the horizontal displacement $x_{EPM}$ and tilting angle $\theta_{EPM}$ of a single EPM and dual EPMs to tilt the CE with a tilting angle $\theta_{desired}$ without any translational motion. To ensure that the tilting motion of the CE occurs without any translational motion, Equations (4) and (6) must be satisfied. Simultaneously, the difference between the desired alignment direction of the CE and the actual CE alignment direction in Equation (9) should be minimized. Because the vertical distance z between the EPM and the CE is determined by the patient’s body size, the control variables are the position and orientation of the EPM. We propose an optimal control problem of EPM to generate the tilting motion of the CE without any translational motion as follows:(10)$$\mathrm{Minimize}\;\theta_{error}\;\mathrm{subject}\;\mathrm{to}\;F_{z}\geq F_{g}\;\mathrm{and}\;{\mu_{s}(F}_{z}-F_{g})-F_{x}>0\;\mathrm{to}\;\mathrm{find}\;\theta_{EPM1},\;\theta_{EPM2},\;\cdots ,\;\theta_{EPMn},\;x_{EPM1},\;x_{EPM2},\cdots ,\;\mathrm{and}\;x_{EPMn}$$

The optimal design problem was solved using the global optimization function ga in MATLAB (2023a).

Once the CE is tilted, the 360° rotational motion of the CE with respect to the contact point between the CE and the stomach is required to scan the lower part of the stomach. Figure 5a shows the trajectory of EPMs used to generate rotational motion of the CE without any translational motion, and it shows that the axis of rotation is the z-axis through the contact point of the CE. During the rotational motion, the CE maintains its tilting position without any translation or downward movement because Equations (4) and (6) have been satisfied. Figure 5b shows the trajectory of the magnetic moment m of the EPM, the unit vector of the magnetic moment N, $\theta_{EPM}$ representing the angle between N and the z-axis, and $\varphi_{EPM}$ denoting the angle between the projection of N to the xy-plane and the x-axis. By controlling the position and orientation of each EPM to the optimized values of $x_{EPM}$ and $\theta_{EPM}$ derived from the optimization problem and varying $\varphi_{EPM}$ from 0° to 360°, rotational motion of the CE can be generated.

### 2.2. Optimal Design of MNS with Dual EPMs

When using a single EPM to control the tilting motion of a CE, we can determine $x_{EPM}$ and $\theta_{EPM}$ to minimize $\theta_{error}$ through Equation (10). However, there might be cases where the optimal solution of $x_{EPM}$ is quite large. This results in a long travel distance of the EPM and consequent increase in examination time. In a dual-EPM system, it is possible to control both $x_{EPM}$ and $\theta_{EPM}$ of each EPM individually. This means that there are more solutions of $x_{EPM}$ and $\theta_{EPM}$ of each EPM to minimize $\theta_{error}$ than in the single-EPM case. Dual EPMs might have a small $x_{EPM1},$ $x_{EPM2}$, $\theta_{EPM1}$, and $\theta_{EPM2}$ to reduce the examination time. We propose a dual-EPM system where EPMs are positioned above and below the CE. As shown in Figure 6a, when the magnetization direction of the CE is along the -z direction and the lens is facing downward, the EPM1 located above the CE and magnetized toward the -z direction generates an attractive force $F_{z,EPM1}$, and the EPM2 located below the CE and magnetized toward the +z direction generates a repulsive force $F_{z,EPM2}.$ The total vertical magnetic force, which is the sum of $F_{z,EPM1}$ and $F_{z,EPM2}$, increases the reaction force between the CE and the contact surface and the friction force consequently. If the friction force applied to the CE is greater than the horizontal magnetic force, translational motion does not occur during the tilting motion of the CE. We can increase the volume of EPM2 until the magnetic torque generated by EPM2 is smaller than that generated by EPM1. Otherwise, the CE may turn over, as shown in Figure 6b. We assume that each EPM has the same diameter and height, and the total mass can be represented as follows:(11)$$M_{total}=M_{EPM1}+M_{EPM2}=\frac{\pi }{4}\rho ({D_{EPM1}}^{3}+{D_{EPM2}}^{3})$$
where ρ, $D_{EPM1}$ and $D_{EPM2}$ represent the density of EPM and the diameters of EPM1 and EPM2, respectively. The EPMs are composed of NdFeB with a residual flux density $B_{r}$ of 1.05 T and density of $7.5\;g/{\mathrm{cm}}^{3}.$ Finally, we formulated an optimization problem as follows:(12)$$\mathrm{Minimize}\;M_{total}\;\mathrm{subject}\;\mathrm{to}\;F_{z,EPM1}+F_{z,EPM2}\geq F_{g}\;\mathrm{and}\;T_{EPM1}\geq T_{EPM2}\;\mathrm{to}\;\mathrm{find}\;D_{EPM1}\;\mathrm{and}\;D_{EPM2}$$

The first constraint in Equation (12), that the sum of the vertical magnetic forces applied to the CE must be greater than the weight of the CE, ensures that the CE is in contact with the upper part of the stomach. The second constraint, that the magnetic torque applied by EPM1 must be greater than that applied by EPM2, ensures that the magnetization direction of the PM of the CE must align with the magnetization direction of EPM1.

Figure 7 shows the combinations of EPM1 and EPM2 that satisfy the constraints within the diameter range from 0 to 240 mm for each EPM. When the respective diameters of EPM1 and EPM2 are 150 mm and 80 mm, the total weight of the EPMs is minimized at 22.59 kg. For a single EPM, when the diameter of EPM2 is zero, the diameter of EPM1 is 240 mm and the weight of a single EPM is 80.34 kg. The proposed dual EPMs reduce the weight of the EPM by 72%.

## 3. Results

### 3.1. Translational Motion of the CE

First, we investigated the translational motion of the CEs with single and dual EPMs, as described in the Materials and Methods section. We used a commercial CE (MiroCam™ from Intromedic, Seoul, South Korea) for verification of the proposed control method. Its weight is 24 mN, and it has a cylindrical PM with a diameter of 9 mm, a height of 2 mm, and a residual flux density of 1.45 T. Its CMOS image sensor captures images with a frame rate of 3 FPS for 12 h. It has a viewing angle of 150°, and it can capture all images of the total viewing angle of 180° once it has a tilting motion of ±15°.

Figure 8 shows the static friction force $F_{f}$ and the horizontal magnetic force $F_{x}$ generated by changing $x_{EPM}$ for the single EPM and dual EPMs. For the single EPM, $\theta_{EPM}$ was set to 90° and the vertical distance between the CE and the EPM was assumed to be 200 mm. For the dual EPMs, $\theta_{EPM1}$ and $\theta_{EPM2}$ were set to be 90° and 270°, and the distance between the CE and EPM1 and the distance between the CE and EPM2 were assumed to be 200 mm, respectively. $x_{EPM1}$ and $x_{EPM2}$ were set to the same magnitude. The horizontal magnetic force and frictional force applied to CE were calculated using Equations (2) and (5) by increasing $x_{EPM}$, $x_{EPM1}$ and $x_{EPM2}$ from 0 to 100 mm with an increment of 0.1 mm. The static friction coefficient $\mu_{s}$ was 0.113, which was measured between the acrylic tank and the CE in the experiment described in Appendix B. When the horizontal magnetic force is greater than the static friction force, translational motion of the CE occurs. When using a single EPM, $F_{f}$ and $F_{x}$ were calculated to be 103.65 mN when $x_{EPM}$ was 9.4 mm. In the case of dual EPM, $F_{f}$ and $F_{x}$ were calculated to be 105.5 mN when $x_{EPM1}$ and $x_{EPM2}$ was 10.1 mm. Therefore, both a single EPM and dual EPMs can generate translational motion of the CE when the EPM moves horizontally by 9.4 mm and 10.1 mm, respectively.

### 3.2. Tilting Motion of a CE

Table 1 shows the simulated horizontal displacement $x_{EPM}$ of the single EPM and the $x_{EPM1}$, $x_{EPM2}$ of the dual EPMs to tilt the CE 80°, 70°, and 60° without any translation according to the vertical displacement. It shows that the horizontal displacement of the dual EPMs is much smaller than that of the single EPM. In Table 1, α is the average reduction rate of $x_{EPM1}$ and $x_{EPM2}$ with respect to $x_{EPM}$, and it increases as the tilting angle increases. It also shows that the dual EPMs can tilt the CE to 70° and 60° where the vertical distance z is 300 mm. In that position, the single EPM can only tilt the CE up to 80°. The proposed dual-EPM configuration reduces the horizontal displacement of the EPMs for the tilting motion of the CE, thus decreasing the movement time of the EPMs and reducing the examination time. Additionally, it allows the tilting control of the CE over a relatively wide range of z values, allowing for the examination of patients with diverse body sizes.

### 3.3. Experimental Verification

#### 3.3.1. Tilting and Rotational Motion Experiment

We developed the Robotically Assisted Magnetic Navigation System for pan-gastrointestinal Capsule endoscopy (C-RAMAN system) to verify the proposed dual EPMs. Figure 9 shows the C-RAMAN system, whose linear and angular robots allow each EPM to move along the x, y, and z directions and rotate with respect to the θ and φ directions. As described in Section III, EPM1 and EPM2 have respective diameters of 150 mm and 80 mm, and they are made of NdFeB with a residual flux density of 1.05 T.

Figure 10 shows images of the CE with the tilting angles of 80°, 70°, and 60°, without any translation motion at the acrylic tank. According to Sliker et al., the friction coefficient of the pig stomach is 0.347, which is 3.07 times larger than the friction coefficient of the watery acrylic board [45]. The large friction coefficient reduces the possibility of undesired translational motion while generating tilting motion. In addition, the stomach near the CE is deformed by the vertical magnetic force, and the dented part of the stomach near the CE can fix the CE. For this reason, the experiment in the acrylic tank is a harsher environment than an actual stomach regarding undesired translational motion. We expect that if the experiment was successful in this environment, the undesired translational motion of the CE would not occur while generating tilting motion in a stomach. Table 2 shows the simulated control variables $\theta_{EPM1}$, $x_{EPM1}$, $\theta_{EPM2}$, and $x_{EPM2}$, which were obtained by solving the proposed optimal design problem. Figure 10 shows that the tilting motions of the CE without any translation motion are effectively controlled after applying the solution of the proposed optimal design problem. Figure 11 shows that the rotational motion of CE is effectively controlled without translation after applying the solution of the proposed optimal design problem. Figure 11a shows the experimental setup for generating the rotational motion of CE, and Figure 11b–d are the images of the CE rotating without translation at the tilting angles of 80°, 70°, and 60°, with the increment of the rotating angle of 120° around the z-axis, respectively. As explained in Section 2.1.3, once the CE is tilted to a certain angle, translational motion does not occur unless the relative distance and orientation of EPM1 and EPM2 are changed. If EPM1 and EPM2 are rotated with a circular orbit centered around the z-axis with radii of $x_{EPM1}$ and $x_{EPM2}$, while maintaining $\theta_{EPM1}$ and $\theta_{EPM2}$ as shown in Table 2, the rotational motion of the CE is generated without any translation motion.

#### 3.3.2. Verification with a Mimetic Stomach

We operated the CE in a mimetic stomach to verify the medical effectiveness of the proposed method. We conducted an experiment using a 3D-printed mimetic stomach, a light shielding tank, and a real-time viewer (RTV) device manufactured by the same company as the commercial CE mentioned in Section 3.1. Figure 12 shows the equipment used in the experiment, and markers were attached to the mimetic stomach at the main areas to be checked during the examination.

Figure 13 shows captured images of the RTV device. We verified that the proposed method could generate tilting motion and rotational motion well without any unintended horizontal movement of the CE and show that it can be used to image all major examination areas in the stomach. This demonstrates that the proposed method and the dual EPM system have medical effectiveness. Figure 13 was videotaped and is attached to this article.

## 4. Conclusions

We have proposed a method to control the motion of a CE in the stomach utilizing a single EPM and dual EPMs for use in the examination of the upper GI tract. We analyzed the motion of the CE by calculating the magnetic torque and magnetic force applied using the point-dipole approximation model. Utilizing the proposed model, we proposed a method to determine the optimal position and orientation of the EPM to generate tilting and rotational motion without any translational motion of the CE. Furthermore, we optimized the weight of each EPM to develop a lightweight MNS. We prototyped the proposed MNS and experimentally verified that it can effectively generate tilting and rotational motions of the CE without any translational motion. The proposed method is expected to contribute to pan-gastrointestinal examination using a CE by controlling the motion of the CE in a stomach accurately. In future studies, animal experiments and clinical trials will be performed to verify the clinical efficiency and safety of this method and system.

## Figures and Tables

**Figure 1 micromachines-15-01032-f001:**
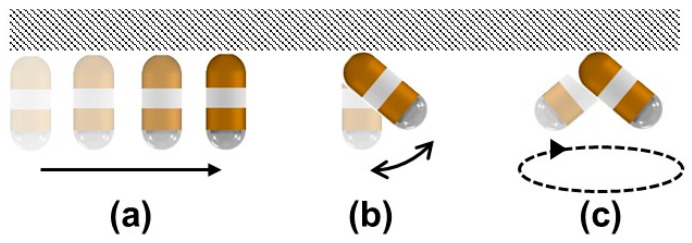
(**a**) Translational motion, (**b**) tilting motion, and (**c**) rotational motion of a capsule endoscope.

**Figure 2 micromachines-15-01032-f002:**
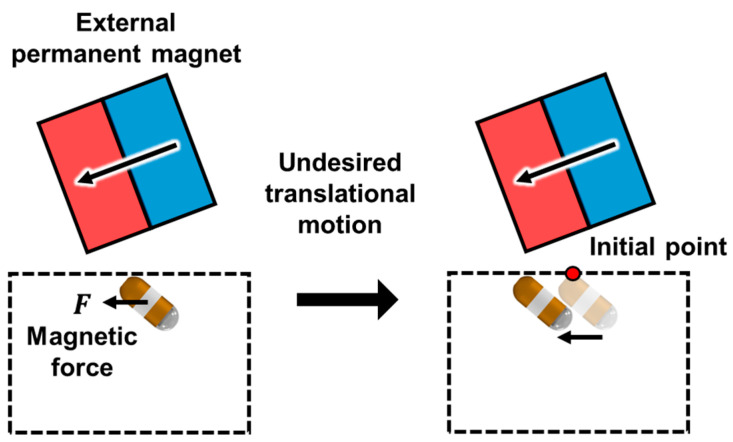
Undesired translational motion occurring when tilting the CE.

**Figure 3 micromachines-15-01032-f003:**
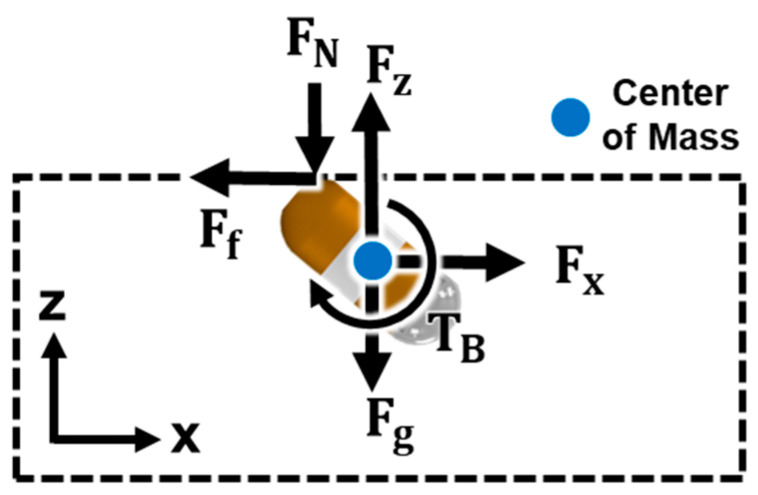
Free body diagram of the CE.

**Figure 4 micromachines-15-01032-f004:**
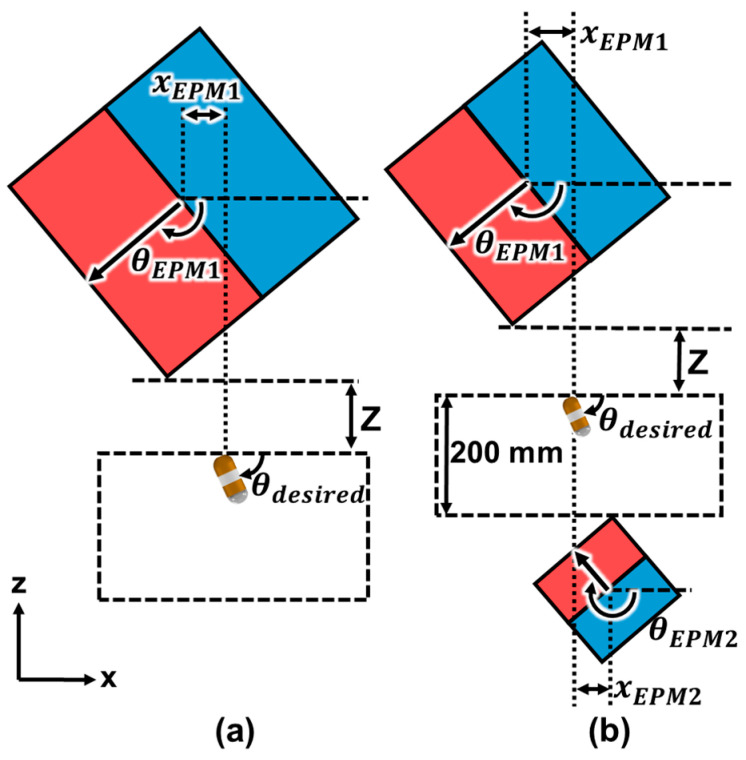
Horizontal displacement and tilting angle of each EPM to generate tilting motion of the capsule endoscope by (**a**) a single EPM and (**b**) dual EPMs.

**Figure 5 micromachines-15-01032-f005:**
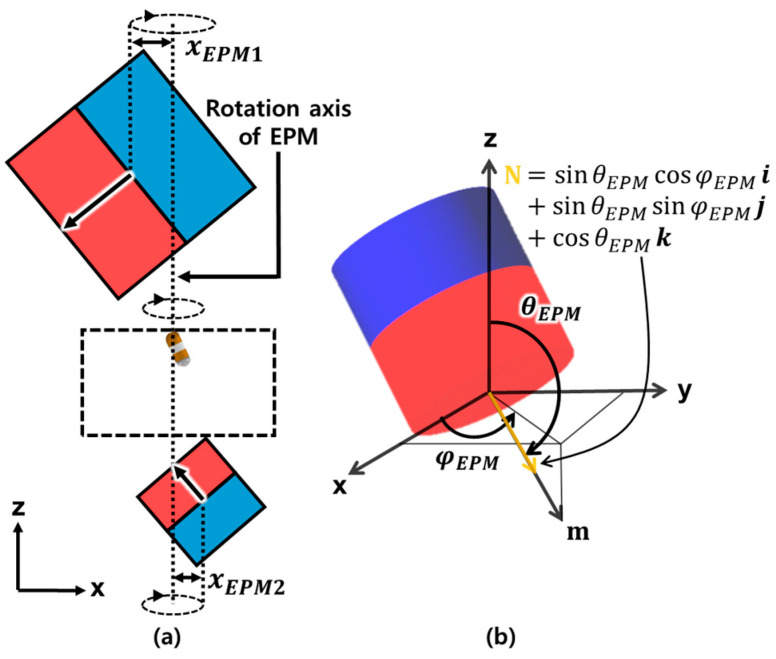
(**a**) Trajectory of EPMs to generate rotational motion without unnecessary translational motion of the CE. (**b**) The magnetic moment m and the unit vector of the magnetic moment N representing the magnetic dipole moment of the EPM.

**Figure 6 micromachines-15-01032-f006:**
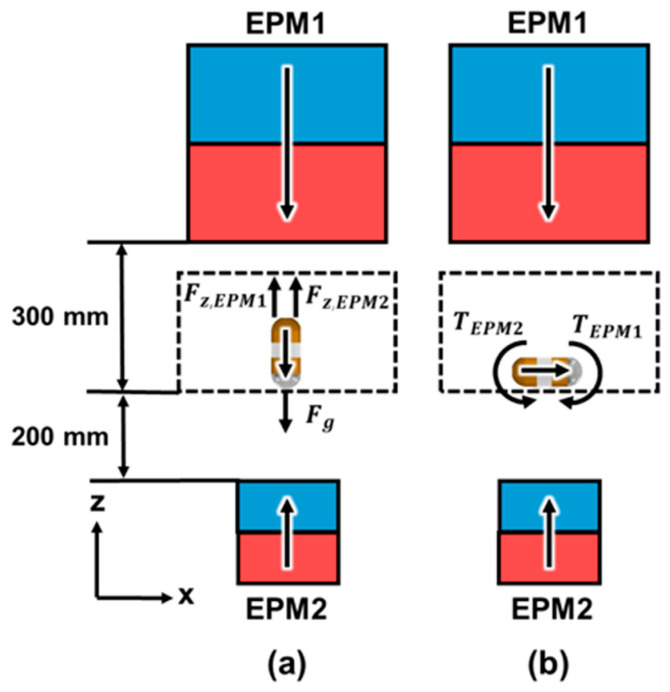
(**a**) A vertical magnetic force and (**b**) magnetic torque applied to the CE.

**Figure 7 micromachines-15-01032-f007:**
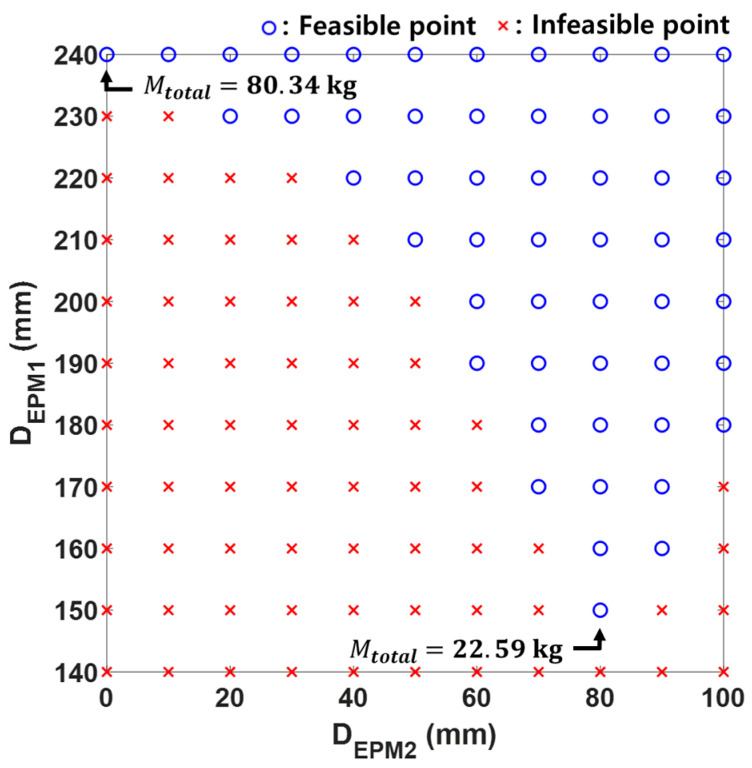
Feasible diameter and length of each EPM satisfying the constraints of Equation (12).

**Figure 8 micromachines-15-01032-f008:**
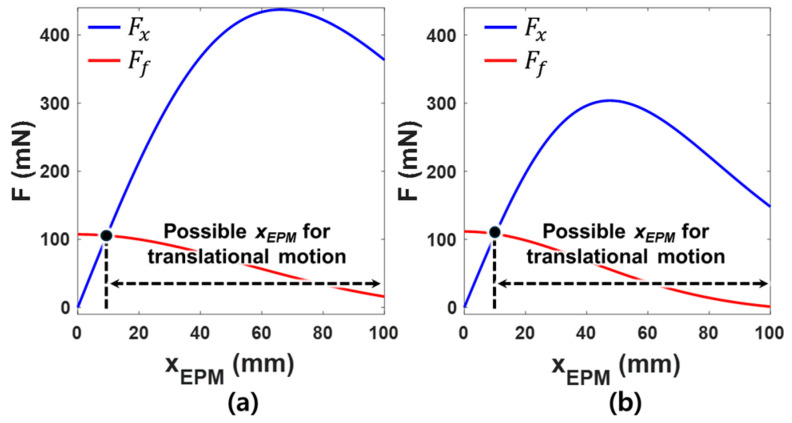
The horizontal magnetic force and maximum static frictional force applied to the CE according to the horizontal displacement of the EPM for (**a**) a single EPM and (**b**) dual EPMs.

**Figure 9 micromachines-15-01032-f009:**
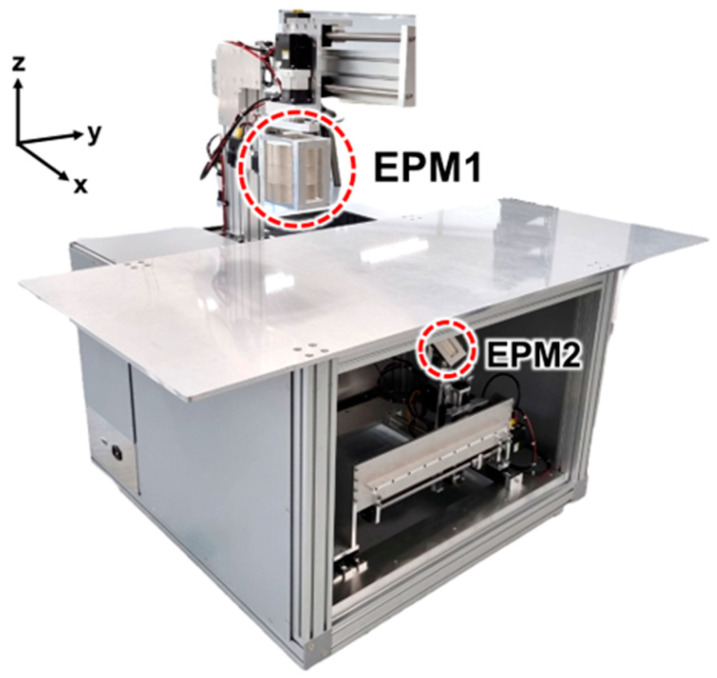
Developed C-RAMAN system with dual EPMs.

**Figure 10 micromachines-15-01032-f010:**
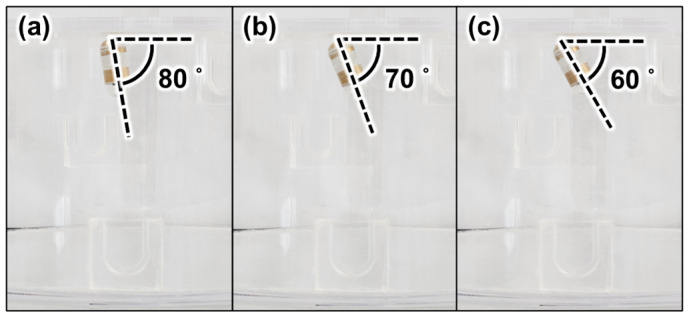
Images of the CE with tilting angles of (**a**) 80°, (**b**) 70°, and (**c**) 60° without any translation motion.

**Figure 11 micromachines-15-01032-f011:**
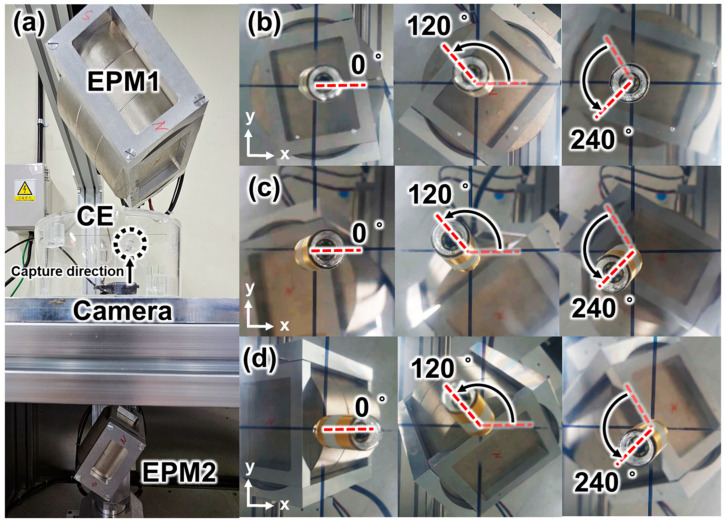
(**a**) Experimental setup to generate rotational motion of a CE using the C-RAMAN system. Captured images of the CE during the rotational motion when the tilting angle of the CE is (**b**) 80°, (**c**) 70°, and (**d**) 60°. (The vertical distance between the CE and EPM1 is 300 mm, and the vertical distance between the CE and EPM2 is 200 mm.).

**Figure 12 micromachines-15-01032-f012:**
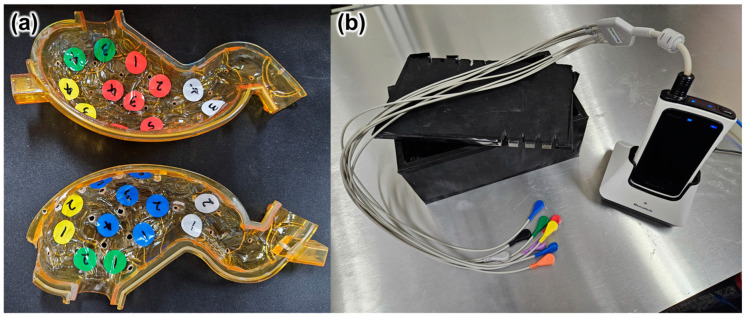
(**a**) Mimetic stomach with markers: yellow-cardia, green-fundus, red and blue-body, white-antrum. (**b**) A light shielding tank and RTV device.

**Figure 13 micromachines-15-01032-f013:**
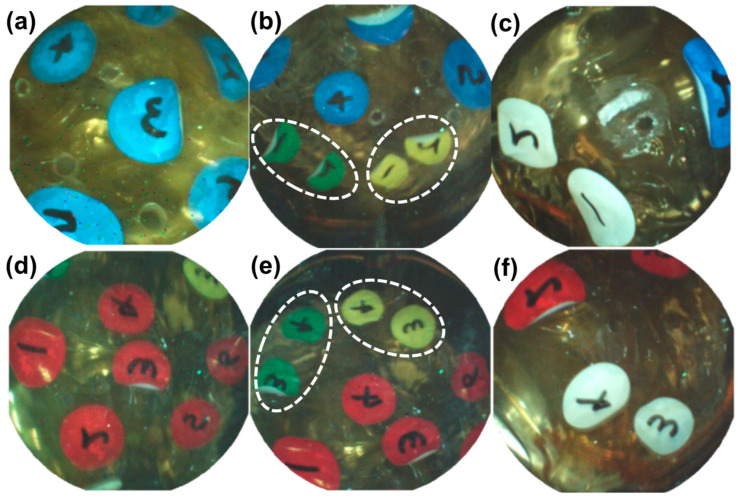
Captured images in RTV verification experiment. (**a**) Upper side of the body (blue marker). (**b**) Upper side of the cardia and fundus (yellow #1, 2 and green #1, 2). (**c**) Upper side of the antrum (white #1, 2). (**d**) Lower side of the body (red marker). (**e**) Lower side of the cardia and fundus (yellow #3, 4 and green #3, 4). (**f**) Lower side of the antrum (white #3, 4).

**Table 1 micromachines-15-01032-t001:** The Horizontal Displacement of the EPM Required to Generate Tilting Motion without Undesired Translational Motion of the CE.

	$\theta_{desired}$ = 80 Deg			$\theta_{desired}$ = 70 Deg			$\theta_{desired}$ = 60 Deg		
	Single	Dual		Single	Dual		Single	Dual	
Z [mm]	$x_{EPM}$ [mm]	$x_{EPM1}$ [mm]	$x_{EPM2}$ [mm]	$x_{EPM}$ [mm]	$x_{EPM1}$ [mm]	$x_{EPM2}$ [mm]	$x_{EPM}$ [mm]	$x_{EPM1}$ [mm]	$x_{EPM2}$ [mm]
200	26	18	9	29	26	18	39	26	18
210	33	28	15	38	29	22	41	29	25
220	44	31	27	58	33	27	64	34	29
230	48	37	32	61	38	33	69	39	33
240	50	38	41	67	39	41	71	41	42
250	53	39	42	73	41	43	77	43	43
260	57	40	42	75	42	44	83	45	46
270	58	42	44	78	44	45	85	46	47
280	59	42	45	84	46	47	91	48	49
290	62	25	45	86	48	49	100	49	51
300	65	45	46	-	50	50	-	51	52
α ^1^ [%]		26	32		38	43		43	47

^1^ $\alpha =\left(x_{EPM1}-x_{EPM}\right)/x_{EPM}*100$ or $\alpha =\left(x_{EPM2}-x_{EPM}\right)/x_{EPM}*100$.

**Table 2 micromachines-15-01032-t002:** Rotational Angle and Horizontal Displacement of Each EPM to Control the CE with Tilting Angles of 80°, 70°, and 60°. (The Vertical Distance Between CE and EPM2 is 200 mm).

$\theta_{desired}$ [deg]	Distance between CE and EPM1 (Z) [mm]	$\theta_{EPM1}$ [deg]	$x_{EPM1}$ [mm]	$\theta_{EPM2}$ [deg]	$x_{EPM2}$ [mm]
80	200	98	18	240	9
	300	95	45	330	46
70	200	114	26	342	18
	300	137	50	310	50
60	200	150	26	342	18
	300	145	51	308	52

## Data Availability

The original contributions presented in the study are included in the article and Appendix A, further inquiries can be directed to the corresponding author.

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
