# Peer review of "Optimal Motion Control of a Capsule Endoscope in the Stomach Utilizing a Magnetic Navigation System with Dual Permanent Magnets"

_micromachines, 2024, doi:10.3390/mi15081032_

Round 1
Reviewer 1 Report
Comments and Suggestions for Authors
The article was carried out at a high scientific level, is devoted to solving an important problem, gastroendoscopy is very important for people and the improvement, and most importantly, the acceleration of this unpleasant procedure is of interest to a wide range of researchers, from physicists and engineers to doctors.
One small drawback, namely: In Figure 5b it is necessary to indicate the vectors m and N.
After eliminating this shortcoming, the article can be published in the Micromachines journal.
Reviewer 2 Report
Comments and Suggestions for Authors
In this manuscript, the motion of the capsule endoscope is controlled using a light-weight magnetic navigational system with dual external permanent magnet. The optimal motion control (position and orientation) is analyzed by point-dipole approximation model. The theoretical calculation is sufficient and detailed. In addition, the authors also carry out experimental verification to generate tilting and rotational motions of capsule endoscope without any translational motion. I would like to recommend the paper for publication in Micromachines.
Minor comment:
1. Please note that the format is uniform: the first letter of the word in the subtitle is capitalized.
2. The experimental results lack comparison with existing reports.
Comments on the Quality of English Languagegood
Reviewer 3 Report
Comments and Suggestions for Authors
The authors (Bae et al.) proposed a method for controlling endoscopic capsules by utilizing single or two-EPMs-coupled systems. They claimed the proposed method could magnetically control the capsules to generate tilting and rotational motions without slipping (so-called “undesired translational motion”). In addition to the capsule control model, the authors optimized weight to minimize the magnetic navigation system. Overall, this work holds a convincing motivation; however, the author failed to achieve their goals due to the confusing research methods. I would like to share my rationale as follows.
[Major Comments]
1. Before getting into the technical details, I would encourage the author to put more effort into maintaining a basic-standard academic professionality. There are self-repetition (Ln 104 vs. Ln 162), incorrect use of symbols (font style, bold, and italic should be used for different purposes), and even undeleted comments (Ln 228-230).
2. I believe mentioning “in the stomach” (e.g., Ln 9) is overclaiming. Given the relatively easy accessibility and large air-laden channels, GI tracts have become one of the main scenarios for biomedical robotic applications. However, due to the mucus and randomly distributed rugae, the stomach environment is significantly different from the acrylic boards that the authors applied in this work. Especially considering capsules’ undesirable slipping under magnetic fields, ex vivo animal stomachs are suggested for a more convincing experimental setup (silicon rubber phantom is also not suggested due to their hyperelasticity).
3. Right following the comment 2, the author should carefully reevaluate the correctness of Eqs. (4) and (6) with the stomach inner wall frictional coef., which should be much lower than that of acrylic (also, the author should explicitly cite the data source of 0.113 in Ln 242).
4. The experimental setup (Fig. 10) for model validation is questionable. The authors should clarify why they applied a flat surface to simulate the upper region of the stomach, and discuss the resulting effects on experimental errors.
[Minor Comments]
1. The introduction section suffers from over-brief descriptions and rude reference citations. For example, “However, these devices were large and complicated, potentially causing injuries to the esophagus and stomach.” How do the authors define the so-called large and complicated? The main text has too many absolute descriptions, which should either be replaced or quantitatively proven.
2. The authors are suggested to add a new figure to explicitly showcase the comparisons between experimental results and model prediction results.
3. A discussion on quantitatively identifying the friction threshold leading to undesirable sliding is suggested to add in the result section (near Fig. 8).
Comments on the Quality of English LanguageFine.
Reviewer 4 Report
Comments and Suggestions for Authors
please see the attachment.

The writing in this article flows well, but it could benefit from some refinement and polishing by a professional editor.
Round 2
Reviewer 3 Report
Comments and Suggestions for Authors
I appreciate the authors effort on addressing my comments.